# Effects of Concentration and Type of Lipids on the Droplet Size, Encapsulation, Colour and Viscosity in the Oil-in-Water Emulsions Stabilised by Rapeseed Protein

**DOI:** 10.3390/foods12122288

**Published:** 2023-06-06

**Authors:** Mirosław M. Kasprzak, Maciej Jarzębski, Wojciech Smułek, Wiktor Berski, Marzena Zając, Karolina Östbring, Cecilia Ahlström, Stanisław Ptasznik, Jacek Domagała

**Affiliations:** 1Department of Animal Product Processing, Faculty of Food Technology, University of Agriculture, 122 Balicka Str., 30-149 Cracow, Poland; marzena.zajac@urk.edu.pl (M.Z.); jacek.domagala@urk.edu.pl (J.D.); 2Department of Physics and Biophysics, Faculty of Food Science and Nutrition, Poznań University of Life Sciences, Wojska Polskiego 38/42, 60-637 Poznań, Poland; maciej.jarzebski@up.poznan.pl; 3Institute of Chemical Technology and Engineering, Poznan University of Technology, Berdychowo 4, 60-695 Poznań, Poland; 4Department of Carbohydrates Technology and Cereals Processing, Faculty of Food Technology, University of Agriculture, 122 Balicka Str., 30-149 Cracow, Poland; wiktor.berski@urk.edu.pl; 5Department of Food Technology, Engineering and Nutrition, Faculty of Engineering, Lund University, P.O. Box 124, SE-221 00 Lund, Sweden; karolina.ostbring@food.lth.se (K.Ö.); cecilia.ahlstrom@food.lth.se (C.A.); 6Lipid Processing Group, The Department of Meat and Fat Technology, Institute of Agricultural and Food Biotechnology, State Research Institute, 4 Jubilerska Str., 04-190 Warsaw, Poland; stanislaw.ptasznik@ibprs.pl

**Keywords:** rapeseed protein, emulsion, microstructure, encapsulation of lipids, partial micro-coalescence

## Abstract

The objective of this study was to extract the rapeseed protein from by-products and further examine the effect of lab-made rapeseed protein on the droplet size, microstructure, colour, encapsulation and apparent viscosity of emulsions. Rapeseed protein-stabilised emulsions with an increasing gradient of milk fat or rapeseed oil (10, 20, 30, 40 and 50%, *v*/*v*) were fabricated using a high shear rate homogenisation. All emulsions showed 100% oil encapsulation for 30 days of storage, irrespective of lipid type and the concentration used. Rapeseed oil emulsions were stable against coalescence, whereas the milk fat emulsion showed a partial micro-coalescence. The apparent viscosity of emulsions raised with increased lipid concentrations. Each of the emulsions showed a shear thinning behaviour, a typical behaviour of non-Newtonian fluids. The average droplet size was raised in milk fat and rapeseed oil emulsions when the concentration of lipids increased. A simple approach to manufacturing stable emulsions offers a feasible hint to convert protein-rich by-products into a valuable carrier of saturated or unsaturated lipids for the design of foods with a targeted lipid profile.

## 1. Introduction

A number of manufactured food products, including yogurts, milkshakes, creams or sauces, are often formulated as emulsions. Thus, the application of emulsion gels is feasible as liquid or solid fat replacers [1,2]. Emulsions are composed of two or more immiscible phases, where one phase is dispersed in the other as small droplets, such as oil droplets-in-water (o/w) or water droplets-in-oil (w/o). In order to prevent the coalescence of emulsion droplets, it is necessary to stabilize them. Amphiphilic molecules, such as small molecular weight surfactants (e.g., monoglycerides, polysorbates, and lecithin) or polymeric emulsifiers (e.g., dairy proteins, modified starches, and celluloses), are commonly added for this purpose [3]. These molecules reduce the interfacial tension between the oil and water phases and increase steric and/or electrostatic repulsion between emulsion droplets, which improves emulsion stability [4].

Proteins are widely used in food emulsion technology, with egg and milk proteins being the most common sources [5]. Among plant-based proteins, soy protein products (e.g., concentrates or isolates) have been extensively utilised as emulsifiers [3]. However, due to allergen concerns and a strong dependence on imported soya, alternative sources of plant-based proteins are being sought by the food industry [6,7]. As a result, other plant-based proteins extracted from corn, pea or rapeseed, have been the focus of increasing research interest [8,9,10]. Rapeseed, which includes Brassica napus, Brassica rapa and Brassica juncea varieties of rapeseed quality, is a significant global crop. In 2022, rapeseed was the second-largest cultivated oilseed (following soybean) in the World (www.statistica.com, access date: 2 June 2023).

Rapeseeds possess significant potential as a source of functional proteins for food applications [1,11]. However, the conventional oilseed valorisation process, which includes defatting, heating, and the use of organic solvents, has a profound impact on both the extractability and functional properties of rapeseed proteins [12,13]. This has resulted in their underutilisation in food applications. In previous research, we demonstrated that the use of an alternative extraction method, such as cold extraction, allows for the preservation of the physicochemical and functional properties of rapeseed proteins [14]. This promising result provides an opportunity to further investigate the functional potential of rapeseed proteins, which could pave the way for their use in a variety of food applications.

Rapeseed protein has a complex protein composition that comprises both storage proteins (cruciferin and napin) and oil body proteins (oleosin and caleosin). Cruciferin and napin (11S globulin and 2S albumin) constitute 60% and 20%, respectively, of the total proteins in mature seeds, and oil body proteins make up the remainder. Cruciferin, with a molecular weight of 230–300 kDa, consists of six subunits, each containing one acidic 30 kDa alpha chain and one basic 20 kDa beta chain, which are linked by a disulphide bond (Akbari and Wu, 2015). Napin is a strongly basic protein with a molecular weight of 12–17 kDa and its well-known that it comprises a 4.5 kDa chain and a 9.5 kDa chain stabilised by disulphide bonds and disulphide bridges [15]. Examining the digestibility of napin and cruciferin among thirteen different rapeseed cultivars, we found that they were all digested at ileal stage, and validated against a proteomic approach [7]. In terms of emulsification properties, cruciferin has been found to possess a notable emulsifying capacity, whereas napin has been reported to negatively impact the emulsifying properties of rapeseed proteins [16]. The emulsifying ability of rapeseed proteins, particularly oleosin, has made them a compelling subject of investigation as an emulsifying agent, thereby positioning rapeseed as a viable, plant-based alternative to other extensively studied protein emulsifiers. In order to further explore the practical application of rapeseed protein, this study aimed to extract rapeseed protein from the by-product of rapeseed after de-oiling. The objective was to investigate the impact of lab-made rapeseed protein on emulsions, specifically focusing on droplet size, microstructure, encapsulation, and viscosity. The emulsions were designed with a concentration gradient of either milk fat or rapeseed oil, and the goal was to compare the effects of rapeseed proteins as stabilizers in emulsions containing saturated (solid) versus unsaturated (liquid) lipids.

To the best of our knowledge, there have been no previous studies comparing the stabilisation properties of rapeseed proteins in emulsions with different lipid types (saturated versus unsaturated). The main aim of this research was to determine whether the content or type of encapsulated lipid would influence the stability and encapsulation properties of the produced emulsions. Ultimately, these findings would contribute to enhancing the value of rapeseed protein concentrate as a lipid carrier, potentially enabling its use as a fat replacer in various applications.

## 2. Materials and Methods

### 2.1. Ingredients

Rapeseed oil (trade name “Kujawski”, Bunge Polska Sp.z o.o., Kruszwica, Poland) and milk fat (trade name “Masło Klarowane”, Mlekovita, Poland) were purchased in a local market. Sodium azide (Sigma Aldrich, Hamburg, Germany) was used as an antimicrobial agent, which was added at 0.02% (*w*/*v*) to the external water phase, prior to a formulation of emulsions. For protein extraction, we used citric acid powder (>95% purity) and NaOH which were purchased from Merck (Darmstadt, Germany) and aspartic acid purchased from Thermo Electron (Milan, Italy). In lipid analysis, we used petroleum ether that was obtained from VWR International (Stockholm, Sweden). Milli-Q water with a conductivity of 0.05 µS/cm^3^ was used in the experiment.

### 2.2. Preparation of Rapeseed Protein Concentrate

Cold-pressed rapeseed press cake (*Brassica napus* L.) used for the extraction of rapeseed protein was a kind gift from Gunnarshögs Jordbruks AB (Hammenhög, Sweden). The proximate analysis of the press cake was 9% moisture, 29% protein, 13% fat, 6% ash and 43% carbohydrates (calculated by differences). The screw press at the Gunnarshög production plant yields cold-pressed oil and the temperature of the oil did not exceed 37 °C, while the press cake had a temperature of about 55–60 °C when exiting the screw press. The press cake was stored at −18 °C prior to the experiments.

The protein was recovered from the press cake on a semi-pilot scale as described previously by Ahlström et al. [17], based on a modification of the method of Wijesundera et al. [10]. The process scheme for the extraction of rapeseed protein from the press-cake was depicted in Figure 1. Two kilograms of press cake was ground in a knife mill (R302 v.v. Robot Coupe, Paris, France) for three minutes to obtain a powder. The milled press cake was dispersed in 18 L tap water in a stirred tank, and the pH was adjusted to 10.5 using 2 M NaOH. The pH was re-adjusted to 10.5 after 10 min. The dispersion was stirred for one hour at 200 rpm using a three-bladed propeller stirrer with a diameter of 140 mm (IKA RW 28 digital, Staufen, Germany). After extraction, the heavy phase was separated from the light phase using a decanter (Decanter centrifuge MD80, Lemitec, Berlin, Germany) at 6687 rpm (2000× *g*) and a differential speed of 10 rpm. The inner diameter of the weir disc was 56 mm. The flow to the decanter was adjusted to 25 L/h with a peristaltic pump (Masterflex Easy-load Model 77200-62, Cole-Parmer, Vernon Hills, IL, USA). The heavy phase containing intact cells, fibres and insoluble proteins was discarded and the light phase containing solubilised proteins was collected. The pH in the light phase was adjusted to pH 3.5 with citric acid powder to induce protein precipitation, and the slurry was separated using a benchtop centrifuge for 20 min at 20 °C at 5000× *g* (Beckman Coulter, Allegra^®^ X-15R Centrifuge, Brea, CA, USA). The precipitate was collected as the final product and the supernatant was discarded. The pH of the precipitate was neutralised to pH 7 with 2 M NaOH using an Ultra Turrax T45 (Janke & Kunkel KG, IKA-WERK, Staufen im Breishau, Germany). NaOH was added in smaller aliquots and mixing was performed for a total of 2 min. Afterwards, the precipitate was freeze-dried (Labconco Lyph Lock 18, Kansas City, MO, USA) for four days. The dried material was then stored in a freezer (−18 °C) until further use.

### 2.3. Proximate Analysis of the Rapeseed Protein Concentrate

All proximate analysis was conducted on freeze-dried rapeseed protein precipitate. The dry matter content was evaluated by drying samples in an oven at 105 °C for 16 h in accordance with the official method of analysis [18]. The protein content was analysed using the Dumas combustion method (Thermo Electron Corp., Flash EA, 1112 Series, Waltham, MA, USA) with a universal conversion factor of 6.25 [19]. The fat content was determined using semi-automatic Soxtec equipment (Tecator AB, Höganäs, Sweden) by solvent extraction with petroleum ether in line with (AOAC, 920.39, 2002). Ash was evaluated according to AOAC 923.03 method [20] using a muffle furnace (B150, Nabertherm GmbH, Lilienthal, Germany). Samples were placed in the oven at 700 °C for 3 h and the samples were allowed to cool in a desiccator for a minimum of 1 h before being reweighed [21]. Each analysis was conducted in triplicates. The carbohydrate content was calculated by difference.

### 2.4. Preparation of Emulsions

A day before an emulsion formulation, a dispersion of 6% rapeseed protein was prepared with 0.02% NaN_3_, followed by stirring at 200 rpm overnight at room temperature. Subsequently, an appropriate proportion of rapeseed protein dispersion and lipid content and type, was prepared and mixed by a high shear mixing at 25,000 rpm for 5 min using a homogeniser (Unidrive x 1000, Ingenieurbüro Cat, Ballrechten-Dottingen, Germany). The lipids were heated up to 40 °C for the process standardisation and also to allow the liquefaction of the solid milk fat prior to the emulsification. Table 1 shows the experimental design of fabricated emulsions. All freshly produced emulsions were subjected to droplet sizing, encapsulation, colour, rheology and spectroscopy analysis.

### 2.5. Characterisation

#### 2.5.1. pH Measurements

The pH was measured on the first day after manufacturing the emulsions using Elmetron CP-505 electrode (Zabrze, Poland). The measurement of each sample was conducted at least 3 times.

#### 2.5.2. Droplet Size

The characteristics of lipid droplet size were determined by a light angle diffraction particle analyser (Analysette 22 Next Nano, Fritsch, Germany). The droplet mean diameters of the emulsion (D[3,2] (Equation (1)), D[4,3] (Equation (2)) were calculated according to the following equations:(1)$$D\left[3,2\right]=\left(\sum n_{i}d_{i}^{3}\right)/\left(\sum n_{i}d_{i}^{2}\right)$$
(2)$$D\left[4,3\right]=\left(\sum n_{i}d_{i}^{4}\right)/\left(\sum n_{i}d_{i}^{3}\right)$$
where d_i_ is the droplet diameter, and n is the number of droplets. The instrument was linked with a liquid dispersion cell operating with water. Two to four droplets of samples were introduced to the water system as indicated by the equipment software.

Moreover, the parameter related to the width of size distribution was calculated according to Equation (3).
(3)$$\mathrm{Span}=\left(d_{\left(90\right)}-{\;d}_{\left(10\right)}\right)/d_{\left(50\right)}$$
where d_(10)_, d_(50)_, d_(90)_ are the equivalent volume diameters at 10%, 50% and 90% of cumulative volume, respectively. Each sample was analysed in four replicates.

#### 2.5.3. Microstructure

The emulsions were visualised using a light microscope (MT5310L, Meiji Techno Co., Ltd., Campbell, CA, USA). A drop of the emulsion was placed onto a glass slide and mixed with a drop of milli-Q water, followed by sliding over a glass cover slip. Microscopic inspection allowed the validation of droplet size and assessment of droplet flocculation. At least ten images were acquired for each of the samples.

#### 2.5.4. Rheology

The apparent viscosity of fabricated emulsions was conducted at a shear rate of 1–300 s^−1^ by Thermo Scientific Haake iQ Rheometer (Massachusetts, MA, USA) equipped with a Haake SC 150-A 10 thermostat. The samples were analysed at 20 °C with geometries such as a cup CCB25 DIN and a rotor CC25 DIN Ti. The obtained flow curves were processed by Ostwald de Waele rheological model (known as the power law model) (4):(4)$$\tau =\;K\cdot \;{\dot{\gamma }}^{n}$$
where: τ—shear stress (Pa), K—consistency coefficient (Pa·s^n^), $\;\dot{\gamma }\;$—shear rate (s^−1^), and n—flow behaviour index.

Apparent viscosity was also reported at a shear rate of 300 s^−1^. Each sample was measured at least three times.

#### 2.5.5. Colour Coordinates

The emulsions were analysed using a spectrophotometer (Konica Minolta CM-3500d, Osaka, Japan) within a day of preparation. Prior to analysis, the calibration was carried out using black and white enamel. The measurements were conducted in the reflectance mode, illuminant D65 and observer angle 10°. The impact of lipid type and content on the overall colour of emulsions was characterised by measuring their L*, a* and b* values. The L* described the lightness of the sample (0 = black, 100 = white), a* ranged from green (−) to red (+) and b* ranged from blue (−) to yellow (+). Each sample was analysed in four replicates.

#### 2.5.6. Centrifugal Encapsulation

The encapsulation efficiency of the emulsion was measured by loss of oil (*LO*) at days 1 and 30 with at least triplicates. A sample of 1.0 g was weighted into an Eppendorf tube (2.0 mL) and centrifuged at 10,000 rpm for 30 min at 4 °C as described by [22]. According to the method, the amount of the expelled oil is removed allowing the weighing of the remaining mass. The encapsulation of oil is calculated as shown by Equation (5) below:(5)$$LO=\frac{\left(m_{i}-m_{f}\right)}{\left(m_{i}-m\right)}\times 100\%$$
where *m_i_* is the sample mass including an Eppendorf tube, *m_f_* is the mass of both sample and Eppendorf after the removal of free oil, and *m* is the Eppendorf mass alone.

#### 2.5.7. FTIR Analysis

The FTIR spectra were registered using a Spectrum Two FT-IR (PerkinElmer, Waltham, MA, USA). The spectrometer was equipped with a Universal ATR module with diamond crystal. The samples were put directly on the crystal with adjusted force gauge c.a. 20. The data were collected during three repetitions in spectral range λ = 500–4000 cm^−1^. 

## 3. Results and Discussion

Stabilisation of lipid-in-water emulsions by rapeseed proteins is well known [10,15,16]. However, there is limited information on whether a concentration or type of lipid could affect the stability of emulsions. In our study, the rapeseed protein concentrate was extracted in a laboratory on a semi-pilot scale. The rapeseed protein concentrate had 50.6% of protein, 19.9% of lipids, 19.2% of carbohydrates and 9.8% ash.

### 3.1. Microstructure and Encapsulation of Emulsions

The stability, sensory characteristics and texture of final products as well as the fate of encapsulated ingredients during storage time, may be affected by the macroscopic properties of emulsions [23,24]. Emulsion droplet size can be influenced by various factors, including emulsifier concentration and type, internal and external phase ratio and homogenisation conditions [25]. In the current study, oil-in-water emulsions were emulsified by rapeseed protein, while commercial rapeseed oil or milk fat was used as an internal phase at concentrations of 10%, 20%, 30%, 40% and 50%. All emulsions had a pH value 6.2 (±0.3). All rapeseed oil emulsions demonstrated no sign of coalescence in the course of a 30-day storage period, regardless of the content of added lipids. However, the milk fat emulsions showed some partial micro-coalescence when the microstructure was examined. However, there was no lipid expelled at the surface of emulsions, irrespectively of milk fat content. Figure 2 and Figure 3 displayed the oil-in-water emulsions stabilised by rapeseed proteins.

Optical microscope observations revealed the presence of flocculation in all produced emulsions. Figure 4 displayed a microstructure example of 20% rapeseed oil and milk fat emulsions. Light scattering measurements of droplet size confirmed the visual inspection, indicating a bi- or polymodal distribution in the tested samples with a wider distribution possibly due to some aggregation (Figure 5 and Figure 6).

In rapeseed oil-based emulsions, the increased concentration of lipids in the formulation led to an increase of droplets in the larger size area (80–110 µm) and a decrease of droplets in the smaller size area (1–10 µm droplets). In our previous study, it was shown that rapeseed protein concentrate in phosphate buffer gave rise to peaks in the size interval of 7–9 µm [26]. We, therefore, suggest that the small peaks of less than 10 µm are aggregated rapeseed proteins that are not needed to cover the emulsion oil-water interface. This is in line with the results presented in Figure 5 and Figure 6. When the oil concentration in the emulsions is low there is a limited oil interface that needs to be covered with protein molecules, hence a large surplus of detached protein molecules in the continuous phase. Further, when the oil concentration is increased, a larger oil interface is created hence a larger proportion of the available rapeseed protein molecules are arranged at the interface to stabilise it. In those emulsions, the proportion of detached protein molecules is lower and the peak in the smaller size range decrease. Span values simultaneously reduced with increased oil concentration in the emulsion formulations (Table 2). Similarly, to rapeseed oil-based emulsions, the milk fat counterparts also showed a reduction in Span values with an increase in lipid content with the exception of the 10% milk fat emulsion. An increase in milk fat content led to a shift in the droplet distribution towards larger droplet sizes (Figure 5). This was due to a lower ratio of protein to lipid in high lipid-concentrated emulsions, allowing a smaller proportion of protein to stabilise the lipid droplets and thereby generate a larger droplet size, as explained by Linke et al. [27].

Overall, the droplet size (both D_4,3_ and D_3,2_) for emulsions with rapeseed oil was smaller than for emulsions with milk fat (Table 2). The differences were more pronounced at higher lipid concentrations in the emulsions. This result could be a function of the protein structure of one of the rapeseed proteins. Oleosin is an oil body protein and its physiological role is to emulsify the lipids inside the seed, resulting in a large interface. This large oil surface allows the plant to access energy rapidly when the environment is suitable for growth. This arrangement with oil stabilised by oil-body proteins, increase the stability against aggregation and coalescence, even when the plant is subjected to temperature and moisture fluctuations [28]. Oleosin has a distinct molecular structure with N-terminal and C-terminal with a scattered distribution of hydrophilic and hydrophobic regions. The hydrophilic regions are facing the aqueous phase whereas the hydrophobic regions are facing the lipid phase. Oleosin also has an unusually long double-bonded hydrophobic region. In fact, this is the longest hydrophobic region found in proteins from organisms. The folding results in a 180° turn anchoring the protein into the oil phase [29]. This arrangement allows the oleosin protein to associate effectively with the oil-water interface. We hypothesize that the reason why rapeseed protein concentrate emulsified rapeseed oil more effectively than milk fat could be that the rapeseed protein structure and folding at the oil-water interface are designed to anchor and stabilize oil with long unsaturated fatty acids, just as the case is in the botanical structure. Further research is needed in order to understand the full mechanism and answer questions if it is the fatty acid length or the degree of unsaturation that is the key factor.

### 3.2. Rheology and Encapsulation

Among the rapeseed oil-based emulsions, the apparent viscosity raised from 0.01 Pa·s in the emulsion with 10% lipids to 0.02, 0.04, 0.10 and 0.30 Pa·s in emulsions with 20, 30, 40 and 50% lipid-rich emulsions, respectively (Table 3). Similarly, with an increase in milk fat, the apparent viscosity increased from 0.02 Pa·s in 10% lipid emulsion to 0.32 Pa·s in 50% lipid emulsions. Irrespective of the type or lipid amount, the apparent viscosity of examined emulsions decreased with a raising shear rate, indicating a shear thinning behaviour. A raising of lipid content led to an increase in the compaction of emulsion droplets (restricting droplet movements), which subsequently enhanced the viscosity of the rapeseed protein stabilised emulsions.

Our results were in line with a study by Zhang et al. [30] who investigated the raising content of soybean oil, olive oil and menhaden oil in the emulsion structure and found that the viscosity of emulsions progressively increased with the increase of oil content. Moreover, a study by Lu et al. [31] also showed that the higher content of coconut oil in the emulsion structure led to a greater viscosity of the emulsion.

The flow properties of samples were described by the Oswald de Waele model. The fittings resulted in R^2^ ≥ 0.99 with the exception of the 40% milk fat-based emulsion having R^2^ = 0.74. As anticipated, the flow behaviour index of samples reduced with increased lipid concentration. The flow behaviour index categorizes into pseudoplastic/shear thinning behaviour as a departure from Newtonian (n close to 1) to non-Newtonian fluid behaviour (n < 1) [32]. The consistency index indicates the viscous nature of the examined emulsion and shows how the consistency of the sample is related [33]. High lipid emulsions had low n values, indicating most likely viscoelastic properties.

The oil relation in the emulsion matrix showed that the produced emulsions had a 100% encapsulation rate at day 30 of storage at 4 °C, irrespective of the type and concentration of lipids. This is due to the mechanism of rapeseed protein stabilisation of the emulsion. Rapeseed protein structures are able to unfold at the oil-in-water interface providing a flexible layer around the lipid droplets [34,35].

### 3.3. Colour Characterisation

The selection of food colour is a crucial element in the decision-making process of food consumers, as it not only impacts the taste threshold, perception of sweetness, food preference, pleasantness and acceptability but also plays a significant role in determining food quality [36]. The colour of the emulsion is contingent upon its scattering and absorption effectiveness. Scattering efficiency primarily relies on droplet characteristics such as size, concentration, aggregation or relative refractive index, whereas the absorption efficiency is chiefly influenced by dye characteristics, including absorption spectra and concentration [37].

The characterisation of the visual appearance of emulsions is depicted in Table 4. Fabricated emulsions had a slight red tinge (a* = 0.5 to 1.8). As they were close to zero, it showed that none of the emulsions had a strong red-green colour. All emulsions had a moderate yellowish colour. Although its intensity slightly reduced with an increase in lipid content, numerical values of b* were in the similar range of 14.0–16.8. The highest lightness was detected in the emulsions with high lipid concentrations, resulting in a L value of 67.2 and 66.5 for rapeseed oil and milk fat emulsions, respectively. Although it has been reported that a reduction in the protein-to-oil ratio and reduced size of lipid droplets might contribute to an increase of the emulsion lightness [38,39], in our study the increasing content of lipids and simultaneously reducing content of rapeseed protein had a greater impact on L value than the size of droplets. This was attributed to the fact that the rapeseed protein concentrate alone had a brownish colour. As a result, some of the pigments from the concentrate had adsorbed light waves, which diminished the fraction of light waves reflected from the emulsion surface. This led to a reduction in emulsion lightness. The brownish colour of rapeseed protein concentrate originated from oxidised phenolic compounds. In rapeseed, the major phenolic compounds are sinapine and condensed tannins [40]. Under alkaline conditions, they undergo enzymatic and non-enzymatic oxidation and form quinones, which react with proteins. These phenol-protein complexes result in a dark green or brown colour of the protein solutions. When the proteins are precipitated, the colour remains, since it cannot be washed from the protein concentrate [41]. The possible concern with colour should be further elaborated within the matrix of new food product design when the inclusion of rapeseed protein stabilised emulsion for the fat reduction strategy is applied.

### 3.4. FTIR—Spectroscopy

Infrared analyses performed on emulsion systems and their components yielded additional information. Both the rapeseed oil and milk fat spectra (Figure 7 and Figure 8) showed signals between 2800 and 3000 cm^−1^ from the stretching vibrations of the C-H bonds in the saturated alkane chains present in the saturated fatty acid residues. Additionally, the saturated fatty acid residues showed a signal from vibrations of carbonyl groups in glyceride esters—at 1720 cm^−1^ in milk fat and at 1740 cm^−1^ in rapeseed oil. At lower wave numbers, the spectra of both lipids contained mainly signals from C-H deformation vibrations and C-O stretching vibrations. The profile of the spectra in this range illustrated the differences in composition between rapeseed and milk fat-rich emulsions. One characteristic difference is the presence of a relatively weak signal of about 3000–3050 cm^−1^ in the rapeseed oil sample, and which is absent in the milk fat sample. This signal can be attributed to stretching vibrations of C-H bonds in the vicinity of C=C bonds present in unsaturated fats.

The spectrum of rapeseed proteins showed signals from the vibrations of the O-H and N-H group bonds (broad band above 3000 cm^−1^), typical of proteins. Signals from C-H and C=O groups were relatively weak and relatively broad. Similarly, signals observed below 1500 cm^−1^ were quite blurred. This indirectly indicates the complexity of the composition of proteins and the diversity of the amino acid units that make them up.

The emulsion spectra are dominated by a broad band between 3000 and 3700 cm^−1^, originating from the stretching vibrations of the O-H bonds in water molecules, further broadened by the formation of hydrogen bonds. Other signals correspond to those for the fat phase and their intensity was proportional to the percentage of fat in the sample. However, this proportionality is stronger for milk fat than for rapeseed oil. Presumably, this is due to the fact that in an emulsion with milk fat, the shift in particle size distribution toward larger particles is more pronounced than for an emulsion with rapeseed oil (see Figure 4 and Figure 5). It is possible that in the FTIR spectra for milk fat, the co-action of the effect of the larger volume contribution of the large droplets and the total larger volume contribution throughout the emulsion occurs.

FTIR spectra at the amide I band between 1700 and 1600 cm^−1^, represent about 80% of the C-O stretching vibration, coupled to the in-plane N-H bending (10%) and C-N stretching (10%) [42]. The amide I region is a diagnostic of protein secondary structure, indicating alpha-helical, beta-pleated sheet, turns and aperiodic conformations [43]. In our study, it seems that the proportion of α-helix, β-sheet, β-turn and random coils might change in emulsion matrices compared to the rapeseed protein concentrate alone. This is possibly due to the unfolding of the protein structure during emulsification [44].

## 4. Conclusions

In this study, we showed the feasibility to formulate rapeseed oil and milk fat-rich emulsions stabilised by rapeseed proteins with lipid concentration gradients. Our findings revealed that rapeseed proteins were more effective in emulsifying rapeseed oil compared to milk fat. The key conclusions of this study are as follows:The examined emulsions remained stable without any oil loss for a period of 30 days at a temperature of 4 °C, regardless of the type and quantity of lipids present.The size of the oil droplets was influenced by the content and type of lipids used in the emulsions.All emulsions exhibited shear thinning behaviour, even though there was a general increase in apparent viscosity with higher lipid content.

By employing a straightforward approach, we were able to manufacture stable emulsions with complete lipid encapsulation. This suggests the potential to transform rapeseed by-products into valuable carriers for both saturated and unsaturated lipids. The resulting emulsions can be utilised as lipid ingredients in the development of novel food products, either as a strategy for quality fat replacements or for reducing fat content.

## Figures and Tables

**Figure 1 foods-12-02288-f001:**
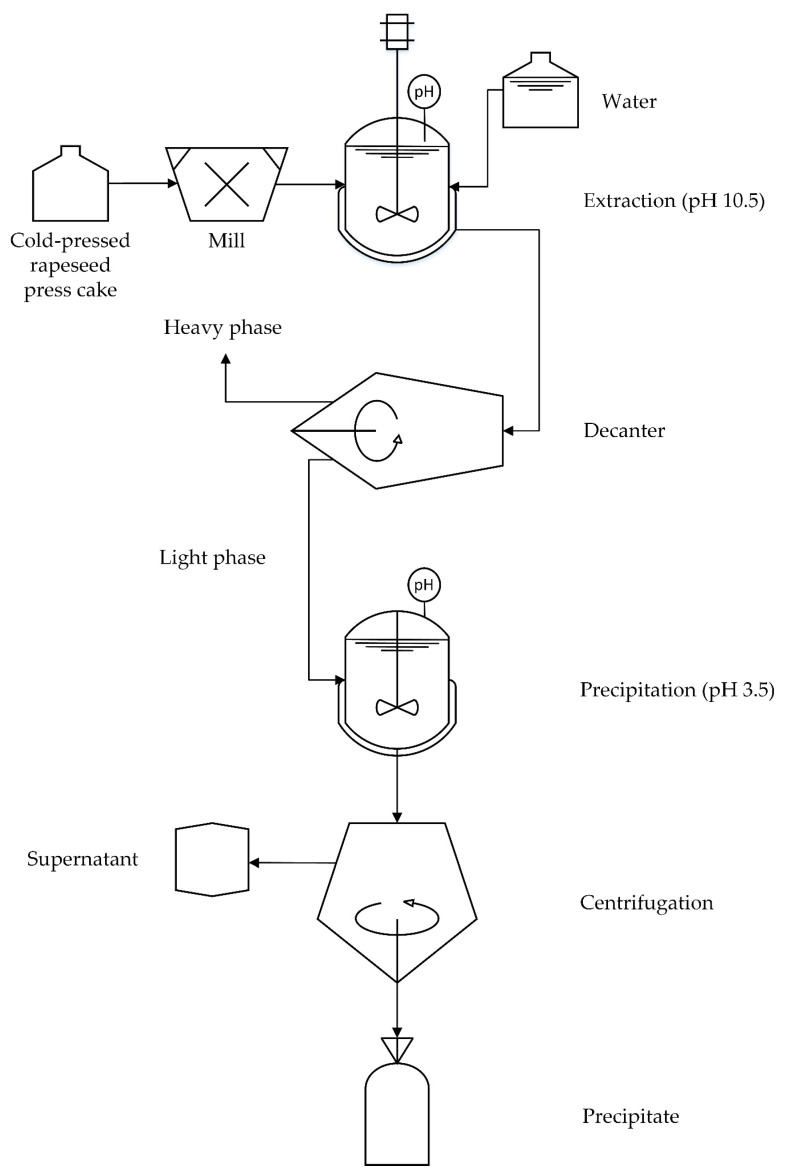
The process scheme for the extraction of rapeseed protein from press-cake.

**Figure 2 foods-12-02288-f002:**
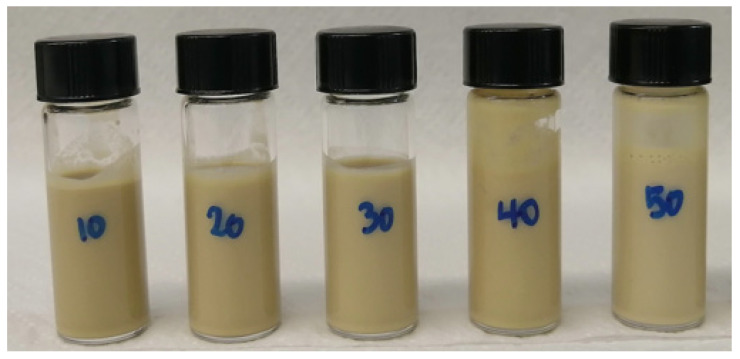
The oil-in-water emulsions stabilised by rapeseed proteins with inclusion of 10, 20, 30, 40 and 50% of milk fat.

**Figure 3 foods-12-02288-f003:**
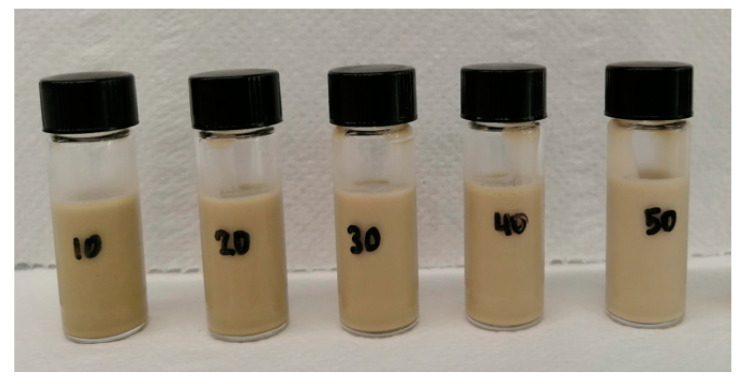
The oil-in-water emulsions stabilised by rapeseed proteins with inclusion of 10, 20, 30, 40 and 50% of rapeseed oil.

**Figure 4 foods-12-02288-f004:**
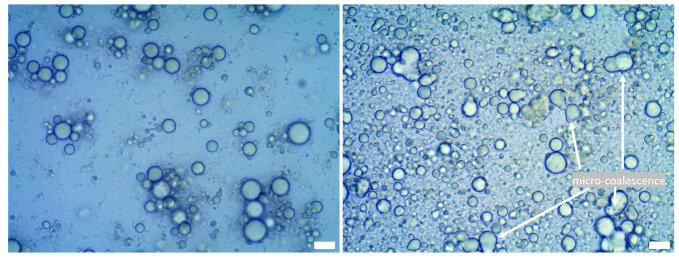
Microstructure of 20% rapeseed oil (**left**) and 20% milk fat rich emulsion (**right**).

**Figure 5 foods-12-02288-f005:**
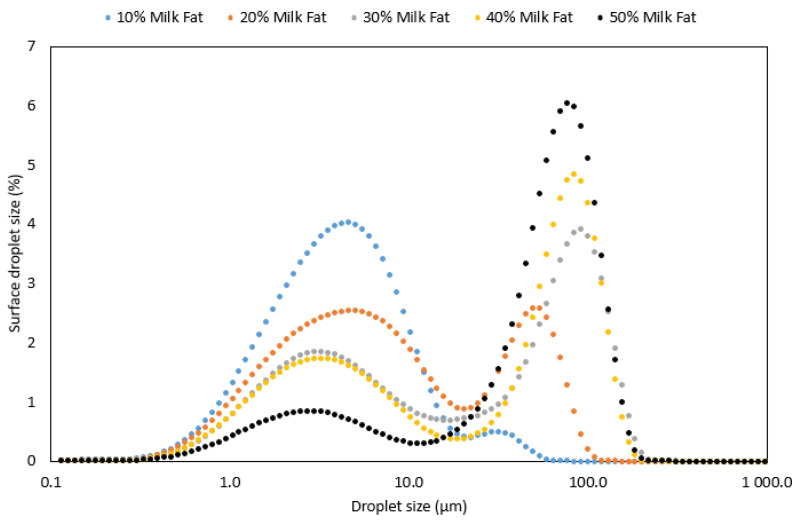
Distribution of droplets in oil-in-water emulsions with inclusion of 10, 20, 30, 40 and 50% milk fat.

**Figure 6 foods-12-02288-f006:**
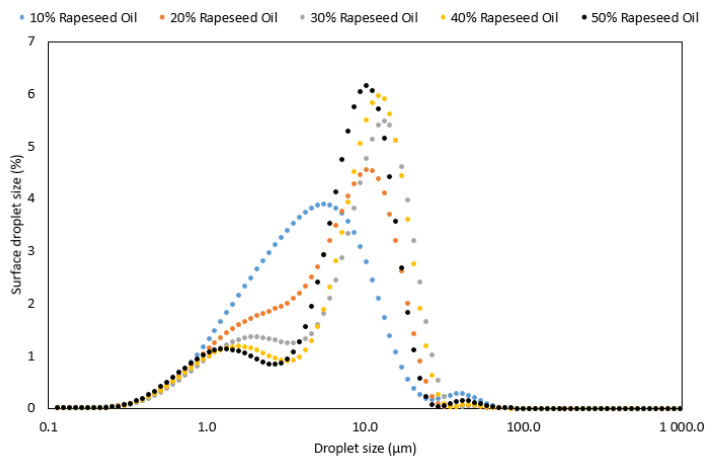
Distribution of droplets in oil-in-water emulsions with inclusion of 10, 20, 30, 40 and 50% of rapeseed oil.

**Figure 7 foods-12-02288-f007:**
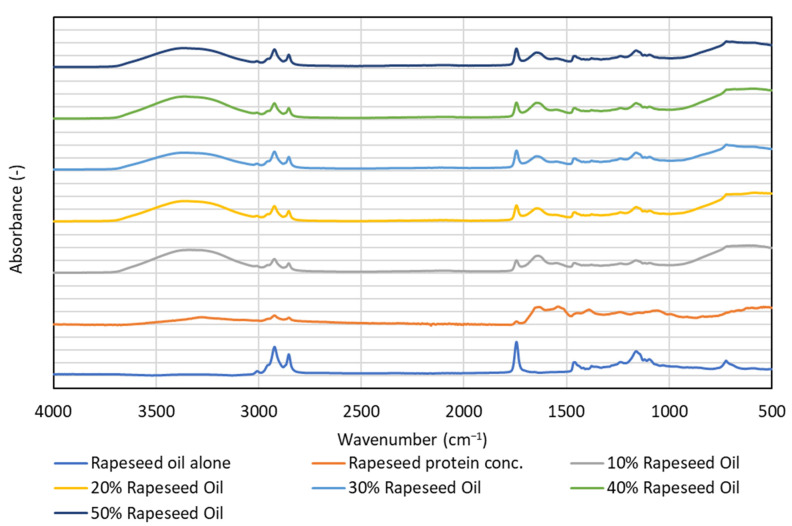
Infrared spectra of oil-in-water emulsions stabilised with rapeseed protein with inclusion of 10, 20, 30, 40 and 50% rapeseed oil.

**Figure 8 foods-12-02288-f008:**
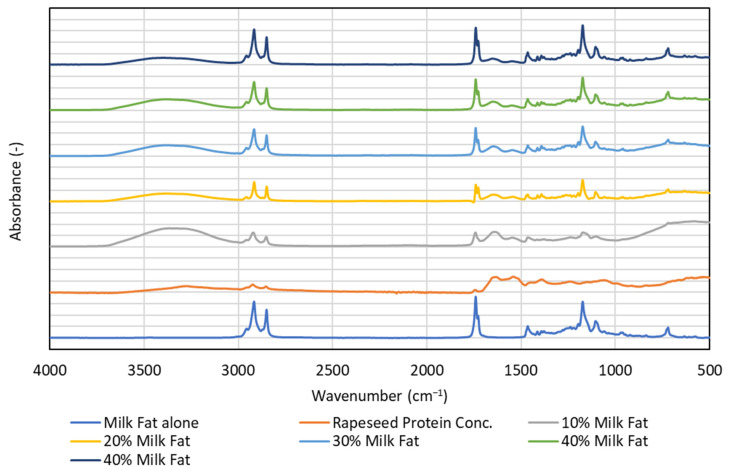
Infrared spectra of oil-in-water emulsions stabilised with rapeseed protein with inclusion of 10, 20, 30, 40 and 50% milk fat.

**Table 1 foods-12-02288-t001:** Experimental design of oil-in-water emulsions stabilised by rapeseed protein.

Sample ID	Type of Lipid	Content of Lipids (%)	Content of Water Phase (%) Enriched with Rapeseed Protein	Protein-to-Lipid Ratio
1	Rapeseed oil	10	90	0.47
2	Rapeseed oil	20	80	0.23
3	Rapeseed oil	30	70	0.13
4	Rapeseed oil	40	60	0.09
5	Rapeseed oil	50	50	0.06
6	Milk fat	10	90	0.47
7	Milk fat	20	80	0.23
8	Milk fat	30	70	0.13
9	Milk fat	40	60	0.09
10	Milk fat	50	50	0.06

**Table 2 foods-12-02288-t002:** Characteristics of oil droplet size in the rapeseed oil or milk fat-rich emulsions.

Sample ID	Type of Lipid	Lipid Content (%)	D_10_ (µm)	D_50_ (µm)	D_90_ (µm)	Span	D_4,3_	D_3,2_
1	Rapeseed oil	10	1.2 (±0.01)	4.3 (±0.04)	11.8 (±0.09)	2.5 (±0.01)	6.0 (±0.04)	2.6 (±0.01)
2	Rapeseed oil	20	1.3 (±0.03)	6.6 (±0.07)	15.5 (±0.10)	2.1 (±0.01)	7.7 (±0.08)	3.1 (±0.05)
3	Rapeseed oil	30	1.4 (±0.07)	9.5 (±0.55)	19.2 (±0.71)	1.9 (±0.04)	10.0 (±0.53)	3.7 (±0.20)
4	Rapeseed oil	40	1.3 (±0.02)	9.5 (±0.03)	18.2 (±0.07)	1.8 (±0.01)	9.8 (±0.03)	3.7 (±0.04)
5	Rapeseed oil	50	1.3 (±0.01)	8.1 (±0.07)	15.3 (±0.12)	1.7 (±0.00)	8.6 (±0.11)	3.5 (±0.04)
6	Milk fat	10	1.2 (±0.01)	4.1 (±0.19)	12.6 (±1.28)	2.7 (±0.19)	6.6 (±0.78)	2.6 (±0.06)
7	Milk fat	20	1.4 (±0.01)	6.6 (±0.04)	54.5 (±1.28)	8.1 (±0.16)	17.9 (±0.42)	3.4 (±0.02)
8	Milk fat	30	1.6 (±0.06)	26.6 (±4.81)	113.6 (±3.80)	4.3 (±0.66)	44.8 (±2.31)	4.5 (±0.18)
9	Milk fat	40	1.7 (±0.01)	41.7 (±1.57)	107.3 (±1.42)	2.5 (±0.07)	46.0 (±0.81)	4.8 (±0.06)
10	Milk fat	50	2.7 (±0.04)	59.1 (±0.53)	111.8 (±0.97)	1.8 (±0.00)	58.9 (±0.61)	8.4 (±0.09)

The values of droplet diameters of emulsions such as D_10_. D_50_. D_90_ are the equivalent volume diameters at 10%, 50% and 90% of cumulative volume. respectively. Results are expressed as mean ± standard deviation.

**Table 3 foods-12-02288-t003:** Rheological properties and encapsulation of emulsions.

Sample ID	Type of Lipid	Content of Lipid (%)	Ostwald de Waele Model			ƞ_(300)_ (Pa·s)	Encapsulation at Day 30 (%)
			K (Pa·s^n^)	n (-)	R2		
1	Rapeseed oil	10	0.02 (±0.01)	0.85 (±0.09)	0.99 (±0.00)	0.01 (±0.00)	100
2	Rapeseed oil	20	0.07 (±0.01)	0.76 (±0.00)	0.99 (±0.00)	0.02 (±0.00)	100
3	Rapeseed oil	30	1.53 (±0.41)	0.38 (±0.04)	1.00 (±0.00)	0.04 (±0.00)	100
4	Rapeseed oil	40	7.10 (±0.66)	0.26 (±0.03)	0.99 (±0.00)	0.10 (±0.01)	100
5	Rapeseed oil	50	47.87 (±2.42)	0.11 (±0.01)	1.00 (±0.00)	0.30 (±0.00)	100
6	Milk fat	10	0.06 (±0.01)	0.79 (±0.03)	0.99 (±0.00)	0.02 (±0.00)	100
7	Milk fat	20	0.33 (±0.24)	0.54 (±0.19)	0.99 (±0.00)	0.02 (±0.00)	100
8	Milk fat	30	1.64 (±0.39)	0.32 (±0.04)	0.98 (±0.00)	0.03 (±0.00)	100
9	Milk fat	40	8.74 (±0.30)	0.27 (±0.07)	0.74 (±0.30)	0.15 (±0.02)	100
10	Milk fat	50	37.55 (±1.10)	0.17 (±0.00)	1.00 (±0.00)	0.32 (±0.00)	100

K—consistency coefficient. n—flow behaviour index. ƞ_(300)_—apparent viscosity of shear rate of 300 s^−1^. Results are expressed as mean ± standard deviation.

**Table 4 foods-12-02288-t004:** Colour characteristics of rapeseed oil and milk fat enriched emulsions.

Sample ID	Type of Lipid	Content of Lipid (%)	L* (D65)	a* (D65)	b* (D65)
1	Rapeseed oil	10	57.9 (±0.21)	1.2 (±0.05)	16.8 (±0.14)
2	Rapeseed oil	20	59.5 (±0.24)	1.4 (±0.05)	16.5 (±0.19)
3	Rapeseed oil	30	61.6 (±0.18)	1.3 (±0.03)	16.1 (±0.19)
4	Rapeseed oil	40	64.0 (±0.06)	1.0 (±0.02)	15.2 (±0.13)
5	Rapeseed oil	50	67.2 (±0.06)	0.5 (±0.01)	14.0 (±0.06)
6	Milk fat	10	56.8 (±0.16)	1.7 (±0.04)	16.5 (±0.17)
7	Milk fat	20	58.9 (±0.16)	1.8 (±0.03)	16.4 (±0.18)
8	Milk fat	30	60.9 (±0.13)	1.6 (±0.04)	16.2 (±0.17)
9	Milk fat	40	63.5 (±0.06)	1.3 (±0.02)	15.3 (±0.09)
10	Milk fat	50	66.5 (±0.07)	0.8 (±0.02)	14.1 (±0.07)

## Data Availability

The data used to support the findings of this study can be made available by the corresponding author upon request.

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
