# Peer review of "Effects of Concentration and Type of Lipids on the Droplet Size, Encapsulation, Colour and Viscosity in the Oil-in-Water Emulsions Stabilised by Rapeseed Protein"

_foods, 2023, doi:10.3390/foods12122288_

Round 1

Reviewer 1 Report

There is no detail on the statistical analysis of the study. Therefore, we have little confidence in the results. Pls, provide details on the stat analysis to help me provide feedback.

Pls, find detailed comments in the attached pdf.

Rampant grammatical errors. Pls check

Author Response

Dear Reviewer,

thank you for your comments and suggestions.

Please find the attached manuscript with changes accordingly.

Kind regards

Miroslaw Kasprzak

Reviewer 2 Report

Abstract:  Please include the numerical values of experimental work in this section. 

Introduction:  Improve the last paragraph of introduction section. Add information about what already being done related to experimental sample of present study and what insist you to perform the present work. This will help to fill the gap. 

Conclusions section must be improved. It must represents the major outcomes of your study. 

Author Response

(The authors gave the same response as above.)

Reviewer 3 Report

Overview and general recommendation:

In the manuscript, the authors make emulsions with gradient percentage of milk fat or rapeseed oil and the microstructure, encapsulation, droplet size, color characterization and apparent viscosity of emulsions are well defined.

I find the paper is organized in a proper way and most of the results are well described. The authors perform background research carefully. And major methods are well described in the manuscript and properly used in the research. I suggest the authors add some discussion about the significance and novelty of this research.

Major comments:

1.      It is easy to adjust the pH of liquid, how can you adjust the pH of the precipitate (I think it is solid). Can you explain this a little bit?

2.      The authors show the properties of emulsions with gradient percentage of milk fat or rapeseed oil but they discuss too little about the future application of the research.

Minor comments

1.      Page9 line263, it should be “the small peaks less than 10 µm are aggregated rapeseed”.

the scientific English is good.

Author Response

(The authors gave the same response as above.)

Round 2

Reviewer 1 Report

The statistical analysis of the study must be included, can't do without it. 

Looks largely fine

Author Response

Reviewer nr 1 also mentioned to include the statistics, that has been corrected.
